# Prostatic urinary tract visualization with super-resolution deep learning models

Takaaki Yoshimura[1,2], Kentaro Nishioka[3], Takayuki Hashimoto[3]*, Takashi Mori[4], Shoki Kogame[5], Kazuya Seki[5], Hiroyuki Sugimori[6,7], Hiroko Yamashina[6], Yusuke Nomura[8,9], Fumi Kato[10], Kohsuke Kudo[11], Shinichi Shimizu[2,3], Hidefumi Aoyama[12]

1 Department of Health Sciences and Technology, Faculty of Health Sciences, Hokkaido University, Sapporo, Japan, 2 Department of Medical Physics, Hokkaido University Hospital, Sapporo, Japan, 3 Department of Radiation Medical Science and Engineering, Faculty of Medicine, Hokkaido University, Sapporo, Japan, 4 Department of Radiation Oncology, Hokkaido University Hospital, Department of Radiation Oncology, Hokkaido University Hospital, Sapporo, Japan, 5 Division of Radiological Science and Technology, Department of Health Sciences, School of Medicine, Hokkaido University, Sapporo, Japan, 6 Department of Biomedical Science and Engineering, Faculty of Health Sciences, Hokkaido University, Sapporo, Japan, 7 Clinical AI Human Resources Development Program, Faculty of Medicine, Hokkaido University, Sapporo, Japan, 8 Department of Radiation oncology, Stanford University, Stanford, CA, United States of America, 9 Global Center for Biomedical Science and Engineering, Faculty of Medicine, Hokkaido University, Sapporo, Japan, 10 Department of Diagnostic and Interventional Radiology, Hokkaido University Hospital, Sapporo, Japan, 11 Department of Diagnostic Imaging, Faculty of Medicine, Hokkaido University, Sapporo, Japan, 12 Department of Radiation Oncology, Faculty of Medicine, Hokkaido University, Sapporo, Japan

* thashimoto@med.hokudai.ac.jp

## Abstract

In urethra-sparing radiation therapy, prostatic urinary tract visualization is important in decreasing the urinary side effect. A methodology has been developed to visualize the prostatic urinary tract using post-urination magnetic resonance imaging (PU-MRI) without a urethral catheter. This study investigated whether the combination of PU-MRI and super-resolution (SR) deep learning models improves the visibility of the prostatic urinary tract. We enrolled 30 patients who had previously undergone real-time-image-gated spot scanning proton therapy by insertion of fiducial markers. PU-MRI was performed using a non-contrast high-resolution two-dimensional T2-weighted turbo spin-echo imaging sequence. Four different SR deep learning models were used: the enhanced deep SR network (EDSR), widely activated SR network (WDSR), SR generative adversarial network (SRGAN), and residual dense network (RDN). The complex wavelet structural similarity index measure (CW-SSIM) was used to quantitatively assess the performance of the proposed SR images compared to PU-MRI. Two radiation oncologists used a 1-to-5 scale to subjectively evaluate the visibility of the prostatic urinary tract. Cohen's weighted kappa (k) was used as a measure of agreement of inter-operator reliability. The mean CW-SSIM in EDSR, WDSR, SRGAN, and RDN was 99.86%, 99.89%, 99.30%, and 99.67%, respectively. The mean prostatic urinary tract visibility scores of the radiation oncologists were 3.70 and 3.53 for PU-MRI (k = 0.93), 3.67 and 2.70 for EDSR (k = 0.89), 3.70 and 2.73 for WDSR (k = 0.88), 3.67 and 2.73 for SRGAN (k = 0.88), and 4.37 and 3.73 for RDN (k = 0.93), respectively. The results suggest that SR

**Funding:** This study was supported in part by Japan Society for the Promotion of Science (JSPS)

KAKENHI (Grant Numbers: JP18K15577 and JP22K15797) and the Grants-in-Aid for Regional R&D Proposal-Based Program from the Northern Advancement Center for Science & Technology of Hokkaido, Japan. The funders had no role in study design, data collection and analysis, decision to publish, or preparation of the manuscript.

**Competing interests:** The authors have declared that no competing interests exist.

images using RDN are similar to the original images, and the SR deep learning models subjectively improve the visibility of the prostatic urinary tract.

## Introduction

Urethra-sparing radiation therapy (USRT), which lowers the urethral dose without compromising the target dose, requires steep dose gradients at the boundary between the urethra and prostate; moreover, it is possible to reduce the risk of urinary side effects associated with urethral radiation, such as urethral stricture, urethritis, and bleeding [1]. The most important USRT technique is the visualization of the prostatic urinary tract. Traditionally, the urethra is identified using a urethral catheter or nickel-titanium stent [2, 3]. However, the urethral position may be displaced by urethral catheter placement [4]. Therefore, urethral visualization using MRI without a urethral catheter has attracted attention in USRT [5–7]. Notably, post-urination magnetic resonance imaging (PU-MRI) has been used to visualize the urethra [8]. However, the visibility of the urethra using MRI is still inferior to that of using a urethral catheter.

High-resolution (HR) MRI provides detailed anatomical information. However, HR-MRI is typically characterized by long scan time, small spatial coverage, and low signal-to-noise ratio. Currently, super-resolution (SR) technique aims to generate a HR image from its degraded low-resolution (LR) image using computer vision tasks. SR with deep learning has demonstrated significant advantages over conventional upscaling methods, such as bicubic interpolation in computer vision [9]. In SR with deep learning, learning methods are classified into two types. Dong et al. first demonstrated the training of an SR convolutional neural network (CNN) using supervised learning technique [10]. There are different types of CNN architectures; for example, Shi et al. demonstrated the SR reconstruction of MRI with a novel residual learning network algorithm [11]. The other type of learning technique is unsupervised learning, such as generative adversarial networks (GAN). These deep learning models provide high-quality, natural-looking images. Sood et al. demonstrated that SR using GAN (SRGAN) could be used for SR imaging in prostate MRI [12]. Also, Chen et al. demonstrated that 3D CNN architecture with GAN-guided training provides sharp SR comparable with the referenced HR images [13]. These SR reconstruction techniques have been applied in brain MRI [14] and MR angiography (MRA) [15, 16]. Whole-heart coronary MRA permits non-invasive detection of coronary artery disease.

MRI of the urethra has been generally performed to detect the urethra's boundary for the purpose of treatment planning. To acurately make out a contour of the urethra on MRI, a deep learning-based urethra segmentation method has been established in MRI-only prostate radiotherapy [17]. However, this study collected annotated data from the catheter- or other foreign object-enhanced MRI, and the anatomical position of the urethra in these images can vary from that in MRI performed without external objects. For collection of annotation data with more accurate anatomical position during treatment, high-visibility MRI acquired under the same condition as USRT is required. Therefore, we hypothesized that the SR image using deep learning models improves the visibility of the urethra while retaining its spatial information, thus providing more accurate urethral contours for USRT treatment planning. This study aimed to evaluate the visibility of the urethra in multiple SRCNN models, which previously provided state-of-the-art performance compared to the original imaging. Although many deep learning-based SR methods have been proposed for MRI segmentation, to the best of our

knowledge, this study is the first to demonstrate the SR method for improvement of urethral visibility in the planning of USRT treatment.

## Materials and methods

### Patients

This retrospective study included a total of 900 images of 30 patients with prostate cancer who had previously undergone real-time-image gated spot-scanning proton therapy (RGPT) using three inserted fiducial markers at our institution from October 2019 to October 2020. This retrospective observational study was approved by the institutional review board (approval No. 018–0221). Written informed consent was obtained from all patients. The characteristics and pretreatment clinical data of the patients are shown in Table 1. The National Comprehensive

**Table 1. Patient characteristics and pretreatment clinical data.**

| | | Number | (%) | Mean | Range | | |
| --- | --- | --- | --- | --- | --- | --- | --- |
| | | | | | Min | - | Max |
| Number of patients | | 30 | 100 | | | | |
| Age | | | | 66.4 | 51.0 | | 80.0 |
| Prescribed dose | 70 Gy (RBE)/ 30 fr | 15 | 50.0 | | | | |
| | 63 Gy (RBE)/ 21 fr | 13 | 43.3 | | | | |
| | 60 Gy (RBE)/ 20 fr | 2 | 6.7 | | | | |
| Risk group (NCCN) | Low | 1 | 3.3 | | | | |
| | Favorable Intermediate | 6 | 20.0 | | | | |
| | Unfavorable Intermediate | 10 | 33.3 | | | | |
| | High | 12 | 40.0 | | | | |
| | Very high | 1 | 3.3 | | | | |
| T stage | T1c | 21 | 70.0 | | | | |
| | T2a | 5 | 16.7 | | | | |
| | T2b | 0 | 0.0 | | | | |
| | T2c | 1 | 3.3 | | | | |
| | T3a | 2 | 6.7 | | | | |
| | T3b | 1 | 3.3 | | | | |
| Initial PSA value | <10 [ng/mL] | 22 | 73.3 | 9.2 | 2.1 | | 39.4 |
| | 10–19.9 [ng/mL] | 6 | 20.0 | | | | |
| | ≥10 [ng/mL] | 2 | 6.7 | | | | |
| Gleason score | 5–6 | 1 | 3.3 | | | | |
| | 7 | 17 | 56.7 | | | | |
| | 8–10 | 12 | 40.0 | | | | |
| Hormonal therapy | - | 18 | 60.0 | | | | |
| | + | 11 | 36.7 | | | | |
| Previous treatment | - | 29 | 96.7 | | | | |
| | + | 1 | 3.3 | | | | |
| Nelaton Catheter size [Fr] | N.A. | 6 | 20.0 | | | | |
| | 10 | 21 | 70.0 | | | | |
| | 12 | 3 | 10.0 | | | | |
| SpaceOAR | - | 15 | 50.0 | | | | |
| | + | 15 | 50.0 | | | | |

Abbreviations: N.A., not applicable; NCCN, National Comprehensive Cancer Network; PSA, prostate specific antigen; RBE, relative biological effectiveness.

Cancer Network Clinical Practice Guidelines in Oncology for prostate cancer categorizes patients into low-risk, favorable and unfavorable intermediate-risk, high-risk, and very high-risk groups [18]. Eighteen patients had received hormonal therapy, and one patient had undergone transurethral resection of the prostate (TURP). A Nelaton catheter was used to visualize the prostatic urinary tract in treatment planning computed tomography (CT). Six patients, including those previously treated with TURP, had difficulty inserting a urethral catheter due to pain during insertion or urinary symptoms. To reduce the risk of gastrointestinal toxicities, 15 patients underwent transperineal insertion of 10 ml of polyethylene glycol gel, SpaceOAR (Augmenix Inc., Waltham, MA), into the Denonvilliers' fascia under transrectal ultrasound guidance [19, 20].

## Image acquisition

Three gold fiducial markers (1.5 mm diameter) were inserted simultaneously with SpaceOAR into the prostate gland for RGPT, one week prior to image acquisition of the treatment planning CT and MRI. In the treatment planning CT image acquisition, patients were placed in supine position and immobilized using a vacuum cushion. We performed CT using Optima CT580W (General Electronic Healthcare, Waukesha, WI) until September 2020 and SOMATOM Confidence (Siemens Healthineers, Forchheim, Germany) thereafter; MRI was performed using a 3.0-Tesla MRI scanner with a 32-channel sensitivity-encoding (SENSE) torso cardiac coil (Achieva TX; Philips Healthcare, Best, The Netherlands). PU-MRI was used to identify the prostatic urinary tract [8]. Details on CT and MRI image acquisition have been reported in previous studies [8]. Briefly, PU-MRI was performed using a non-contrast HR two-dimensional T2-weighted turbo spin-echo imaging sequence within a few minutes after urination. The acquisition parameters for PU-MRI were as follows: resolution = 320×320 matrix, voxel size = $0.5 \times 0.5 \times 2.0$ mm$^3$, field of view = $160 \times 160$ mm$^2$, slices = 30, effective time echo = 80 ms, repetition time = 5093 ms, TSE factor = 9, SENSE P reduction factor = 1.4, gap = 0 mm, and acquisition direction = axial. All acquired images were co-registered with the CT image without using a urethral catheter on MIM ver.7.0.4 (MIM Software, Inc., Cleveland, OH), based on the inserted fiducial markers with rigid registration algorithm.

## Super-resolution deep learning model

In this study, we selected previously reported four deep learning models with excellent performance in SR competition. The first one is an enhanced deep super-resolution network (EDSR), which is the winner of the NITRE 2017 Super-Resolution Challenge [21, 22]. The second one is the wide activation for an efficient and accurate image super-resolution network (WDSR), which won the NITRE 2018 Super-Resolution Challenge (realistic tracks) [23, 24]. The third one is a super-resolution generative adversarial network (SRGAN) [25], while the last one is the residual dense network (RDN) [26]. We used the diverse 2 K resolution RGB images (DIV2K) dataset employed in the NITRE 2017, 2018 Super-Resolution Challenge [9, 22, 24]. Each LR image was obtained from the HR DIV2K image by bicubic downscaling in the image domain. Finally, these SR models with tensorflow version are officially being made publicly unavailable; hence, we used third party source codes for EDSR, WDSR, and SRGAN, which have been open sourced under the Apache 2.0 license (https://github.com/krasserm/super-resolution), and also used the source code for RDN under MIT license (https://github.com/hengchuan/RDN-TensorFlow).

The DIV2K dataset comprises 800 training images, 100 validation images, 100 test images, and bicubic downsampled images. In all SR deep learning models, we train with Adam optimizer [27] by setting $\beta_1 = 0.9$, $\beta_2 = 0.999$, and $\epsilon = 10^{-8}$. The number of epochs was 50; batch

size, 16; and learning rate, $10^{-4}$. The loss functions for the SR deep learning models were L1 loss for EDSR, WDSR, and RDN; and VGG54 content loss for SRGAN. The trainable parameters for each network were 43 M, 0.62 M, 1.55 M, and 22 M for EDSR, WDSR, SRGAN, and RDN, respectively. In this study, these architectures were not modified further and were used as proposed in the respective references [21, 23, 25, 26]. For testing, we use 900 PU-MRI images of 30 patients.

## Data analysis

Although typical MRI images have a grayscale, these SR models are limited to importation in 8-bit RGB color, and the trained data in one channel were concatenated with those in the other two channels to construct complete HR color images. Thus, all PU-MRI image data were converted from 16-bit grayscale digital imaging and communications in medicine (DICOM) files into 8-bit RGB portable network graphics (PNG) files. Next, to leverage the existing SR deep learning models, the imputed PU-MRI images of these SR deep learning models were cropped at the edge from the original 320×320 resolution to 310×295 resolution by specifying the width and height parameters without changing the image center. Overall, as shown in Fig 1, the inputted in-plane PU-MRI images of these SR deep learning models had a resolution of 310×295 for the LR image and 1240×1180 for the output SR image. In this study, all networks operated on only the magnitude images instead of the complex-valued MRI data. All SR images were reconverted from the RGB PNG file to the grayscale DICOM file and imported into MIM. We performed these processes in-house with MATLAB's (MATLAB2019b, The MathWorks, Inc., Natick, MA, USA) "mat2gray" function.

We objectively and subjectively evaluated the SR images using the following two methods. First, the complex wavelet structural similarity index (CW-SSIM) was used for objective evaluation of the original and SR images [28]. The CW-SSIM index is calculated on various subbands of an image in the complex wavelet domain. The results are averaged over the whole image to get a resultant quality measure for the whole images. This approach has been proven to be more robust than the basic structural similarity index (SSIM) for geometric image distortions. The CW-SSIM metric was defined using the following equation (Eq 1):

$$CW - SSIM = \frac{2|\sum_{i=1}^{N} c_{x,i} c_{y,i}^*| + K}{\sum_{i=1}^{N} |c_{x,i}|^2 + \sum_{i=1}^{N} |c_{y,i}|^2 + K} \tag{1}$$

where $c_x = \{c_{x,i}|i = 1, 2, \ldots, N\}$ and $c_y = \{c_{y,i}|i = 1, 2, \ldots, N\}$ are coefficients extracted at the same spatial location, $x$ or $y$, for window $i$ in the same wavelet subbands of the two images over a window of size $N$. $c_y^*$ denotes the complex conjugate of $c_y$, and $K$ is a small positive constant. The CW-SSIM ranges from 0% to 100%, where 100% denotes a perfect similarity between the original image and the SR image.

Next, we evaluated the urethra's visibility score for subjective evaluation of PU-MRI with four upsampled SR images using the SR model for each, with and without the use of a urethral catheter in the treatment planning CT as the reference. Two radiation oncologists, who had over 10 years (KN) and over 15 years (TH) of experience in prostate cancer treatment planning, respectively, assessed all the image sets. All images were anonymized and randomized before image review. A 1-to-5 scale (1 = "urethra not visible" and 5 = "urethra visible along its entire length") was used for subjective evaluation of the visibility of the prostatic urinary tract, as reported by Zakian et al [7].

Bland–Altman plots were generated to evaluate the agreement of the urethra's visibility score between operators in each image set. In the Bland-Altman plot, the horizontal axis shows

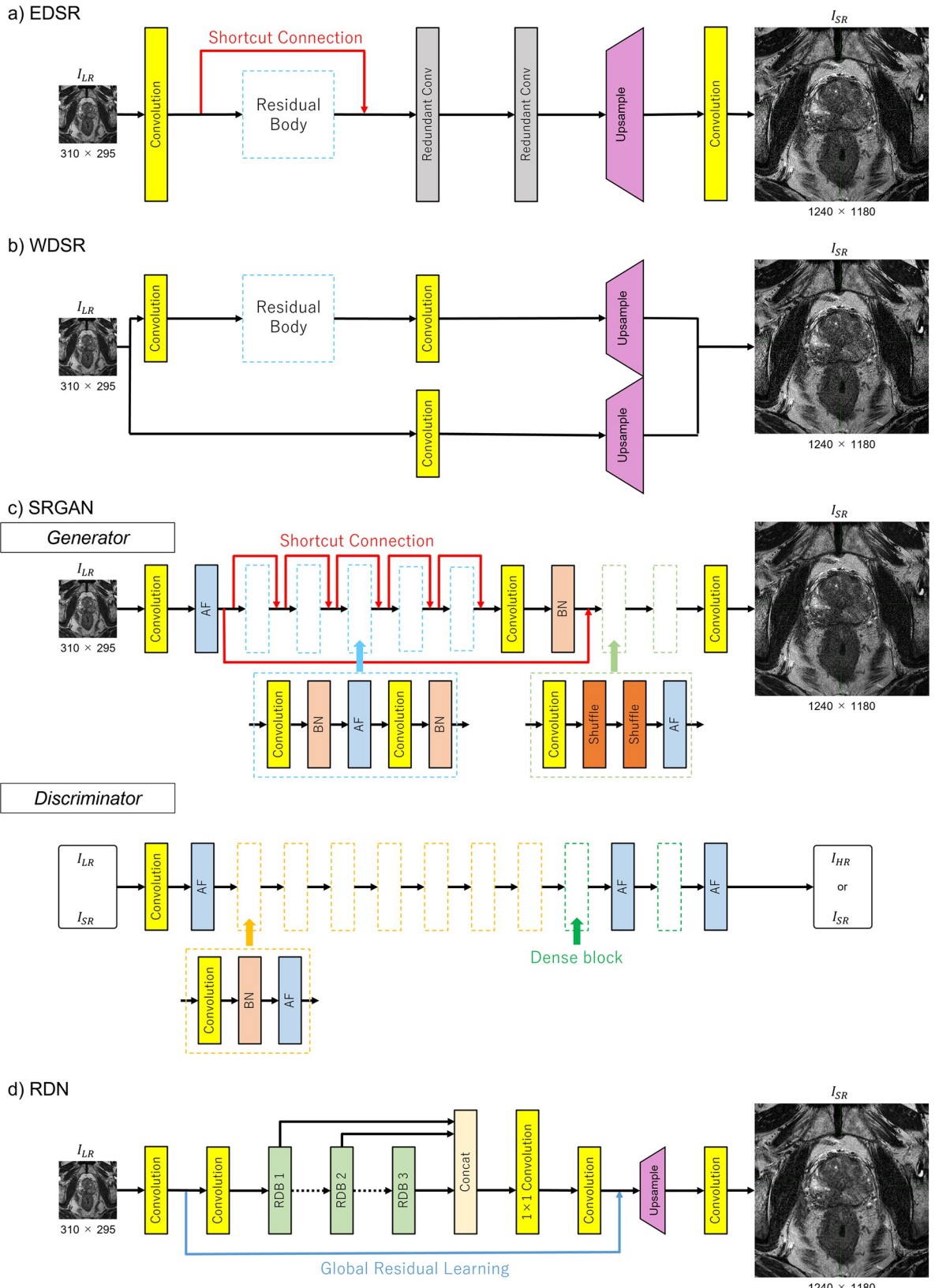

**Fig 1. Architectures in this study.** Input PU-MRI images ($I_{LR}$) were 4 times upscaled by each SR model. (a) EDSR, (b) WDSR, (c) SRGAN, and (d) RDN. Abbreviations: AF, activation function; BN, batch normalization; EDSR, enhanced deep super resolution network; PU-MRI, post urination magnetic resonance imaging; SR, super resolution; SRGAN, photo-realistic single image super resolution using a generative adversarial network; RDB, residual dense block; RDN, residual dense network; WDSR, wide activation for efficient and accurate image super resolution network.

the mean of the urethra's visibility score between operators; the vertical axis represents the difference in the urethra's visibility score between operators. Cohen's weighted kappa (k) was used as a measure of agreement of inter-operator variance [29, 30]. This indicates the magnitude of the disagreement between the operators in the calculation. The interpretation of agreement for k was categorized as follows: poor (k < 0), slight (0 ≤ k ≤ 0.2), fair (0.21 ≤ k ≤ 0.4), moderate (0.41 ≤ k ≤ 0.6), substantial (0.61 ≤ k ≤ 0.80), and almost perfect (k > 0.8).

The paired t-test was used for all statistical comparisons of subjective evaluations between the PU-MRI and SR images in each operator. Statistical significance was set at p <0.05. All statistical analyses were performed using JMP Pro 14 (SAS Institute Inc., Cary, NC, USA).

## Results

In the objective evaluation, the mean CW-SSIM in each SR deep learning model was as follows: 99.86% (95% confidence interval [CI]: 99.85–99.86%) in EDSR, 99.89% (95% CI: 99.88–99.89%) in WDSR, 99.30% (95% CI: 99.29–99.31%) in SRGAN, and 99.67% (95% CI: 99.65–99.69%) in RDN. Fig 2 shows the representative images.

In the subjective evaluation, the mean urethra visibility score for each image by operator 1 was 3.53 (95% CI: 3.16–3.91) in PU-MRI, 3.53 (95% CI: 3.22–3.85) in EDSR, 2.73 (95% CI: 2.42–3.04) in WDSR, 2.73 (95% CI: 2.42–3.04) in SRGAN, and 3.73 (95% CI: 3.38–4.09) in RDN. Also, the mean urethra visibility score for each image by operator 2 was 3.70 (95% CI: 3.46–3.94) in PU-MRI, 3.67 (95% CI: 3.44–3.89) in EDSR, 3.67 (95% CI: 3.42–3.91) in WDSR, 3.70 (95% CI: 3.50–3.90) in SRGAN, and 4.37 (95% CI: 4.14–4.60) in RDN. For operator 1, the urethra visibility score in RDN was significantly higher than that in PU-MRI. However, no

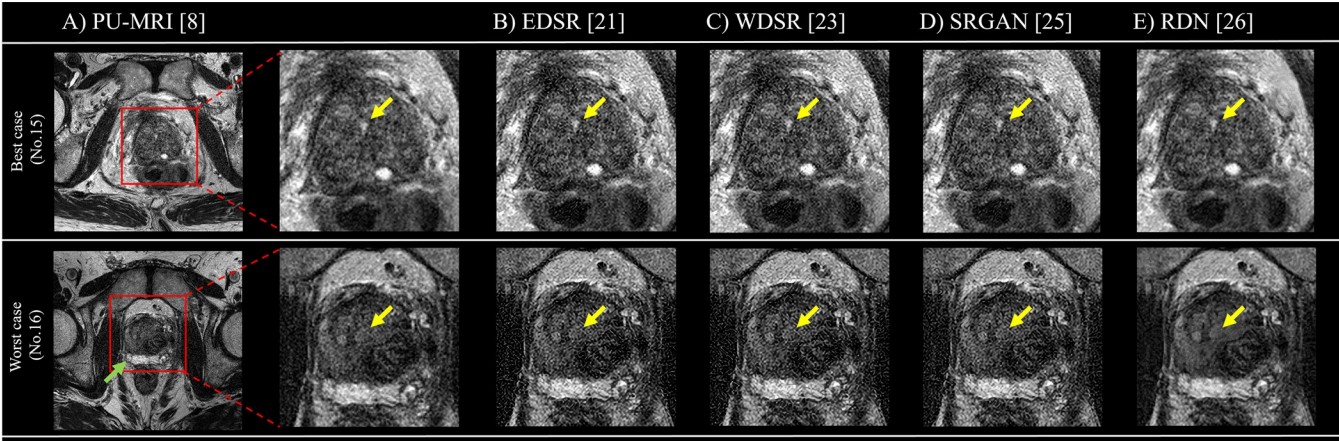

**Fig 2. Representative images identifying the prostatic urinary tract in the axial direction for each SR image.** The above representative images show the best case (No. 15), while the bottom images show the worst case (No. 16) in the subjective evaluation. (A) On the left side of the PU-MRI image is the input image for each SR deep learning model, while on the right side of the PU-MRI image is the enlarged image of the prostate gland and urethra area. The output SR images (B–E) have the same matrix as the magnified PU-MRI image. Yellow arrow shows the prostatic urinary tract. Green arrow shows the gel spacer. Abbreviations: EDSR, enhanced deep super resolution network; PU-MRI, post urination magnetic resonance imaging; RDN, residual dense network; SRGAN, photo-realistic single image super resolution using a generative adversarial network; WDSR, wide activation for efficient and accurate image super resolution network.

significant differences were observed between PU-MRI and other SR deep learning models (EDSR, p = 0.71; WDSR, p = 1.00; and SRGAN, p = 0.77). Despite no significant difference between PU-MRI and RDN, the urethra's visibility score in the other SR deep learning models was significantly lower than that in PU-MRI. Inter-operator agreement for each SR deep learning model was k = 0.93 (PU-MRI), 0.89 (EDSR), 0.88 (WDSR), 0.88 (SRGAN), and 0.93 (RDN).

## Discussion

This study assessed the visibility of the prostatic urinary tract using various SR deep learning methods based on objective and subjective evaluations. The objective evaluation using CW-SSIM showed no clear difference among the SR deep learning models. Moreover, our subjective evaluation found that the mean urethra visibility score in RDN increased in both operators. However, urethral visibility did not increase in the other SR deep learning models. In addition, the results of inter-operator agreements in subjective evaluation of the urethra's visibility by two experienced radiation oncologists were almost perfect. Thus, these results suggest that RDN is the best model for improving the visibility of the prostatic urinary tract among the several SR deep learning models assessed in this study.

In this study, the urethral catheter could not be inserted in six patients during treatment planning CT. Our proposed PU-MRI can visualize the prostatic urinary tract without using a urethral catheter, and is indeed beneficial for patients who cannot undergo urethral catheterization. However, the inter-operator contouring accuracy of the prostatic urinary tract is not high [8]. Notably, when using the SR deep learning model for PU-MRI, the inter-operator accuracy is possibly increased. Since many SR deep learning models have been proposed, we focused on choosing the ideal SR deep learning model in this study.

The peak signal-to-noise ratio or SSIM has been used for the evaluation of various SR models [22, 24, 31, 32], and these indicators can be used to compare images of the same matrix size. To improve the visibility of the prostatic urinary tract for USRT using the SR deep learning model, we performed four times upsampling of the PU-MRI image by SR deep learning models. As a result, the image matrix size between the original PU-MRI and SR images differed, and peak signal-to-noise ratio and SSIM were not used as indicators in this study. Therefore, we used CW-SSIM for the objective evaluation of the original images of PU-MRI and SR images.

This study has some limitations. The first limitation is the explosive spread of the architecture of the deep learning model. Because many architectures are released every year, it is difficult to verify which model is optimal for our purposes [22, 24, 31, 32]. Thus, we selected four SR models that had achieved excellent results in the NTIRE Challenge. The next limitation of this study is the training data set. In this study, we used the DIV2K dataset for training. Most existing SR models are trained and evaluated using simple and uniform degradation datasets, such as DIV2K [9]. Because authentic degradations in real-world LR images are much more complex, it is difficult to evaluate real-world data, especially in medical images. Although the real-world data set was used in recent competitions, medical images were not included [33]. Knoll et al. described the risk of using a database of natural scene images for transfer learning in medical images [34]. However, collecting large amounts of data for training is usually expensive in medical imaging, and it is impossible to acquire HR ground truth images in some cases. Especially, it is difficult to prepare a data set specialized for abdominal MRI. Generally, it takes a long time to acquire HR images using MRI. Indeed, the effect of motion artifacts is increased because of internal organ motion during the long acquisition time. In several previous studies, medical image SR was used to train networks by data acquisition and pre-

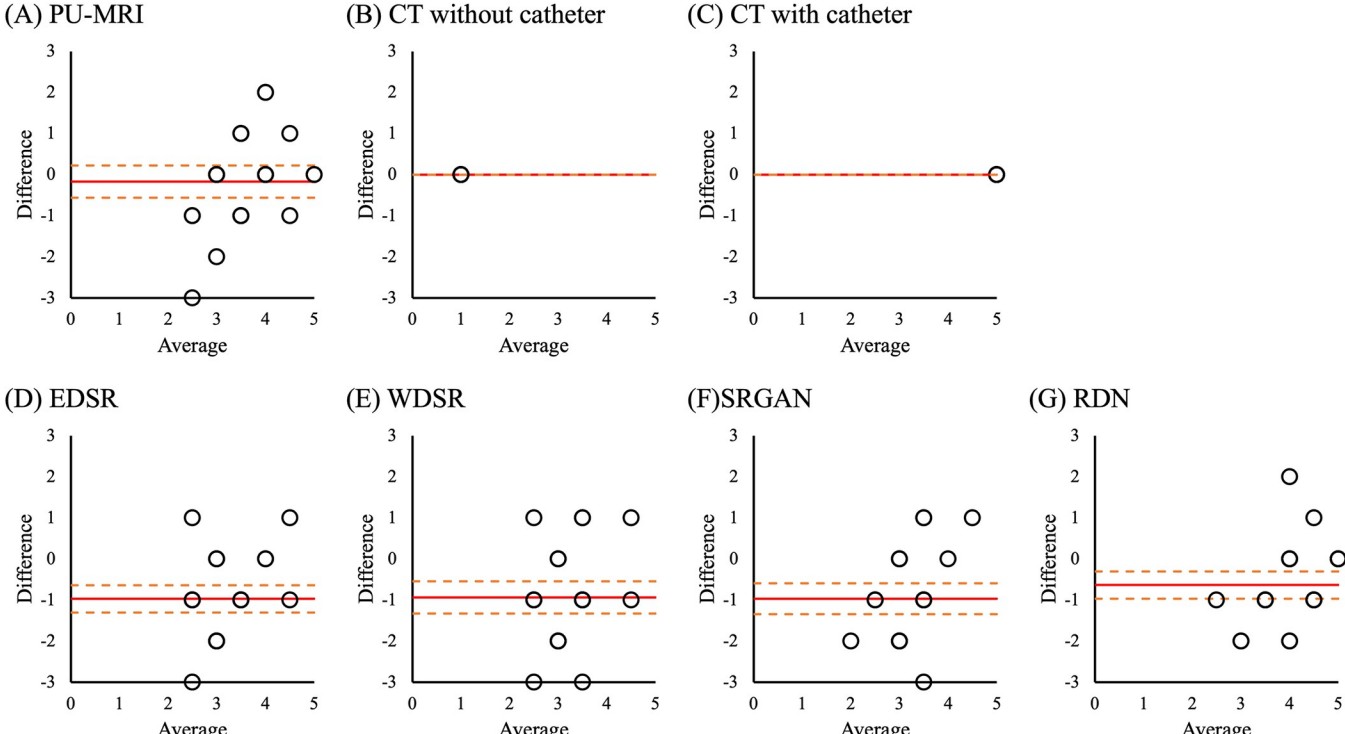

**Fig 3. Bland-Altman plot showing the inter-operator difference in the urethra's visibility score for each image set.** The red line denotes the mean of the difference; the orange dashed line denotes the 95% limits of agreement. Abbreviations: EDSR, enhanced deep super resolution network; PU-MRI, post urination magnetic resonance imaging; RDN, residual dense network; SRGAN, photo-realistic single image super resolution using a generative adversarial network; WDSR, wide activation for efficient and accurate image super resolution network.

processing [11–13]. Thus, in future studies, we need to increase the urethra visibility with SR deep learning models in the training dataset through data acquisition and pre-processing. The other limitation of our study was inter-operator bias, as shown in Fig 3. Although the inter-operator agreement was almost perfect in PU-MRI, the absolute difference in the SR deep learning models was almost 1. This suggested that operator 2 understood the generated SR images more than did operator 1.

## Conclusions

In this study, we objectively and subjectively evaluated the visibility of the prostatic urinary tract on PU-MRI using SR deep learning models. We found that RDN improved the visibility of the prostatic urinary tract while ensuring similarity with the original PU-MRI. The present findings indicate that the SR deep learning model improves the visibility of the prostatic urinary tract without using a urethral catheter.

## Supporting information

**S1 File.**
(DOCX)

## Author Contributions

**Conceptualization:** Takaaki Yoshimura, Kentaro Nishioka, Shinichi Shimizu.

**Data curation:** Takaaki Yoshimura, Takayuki Hashimoto, Shoki Kogame, Kazuya Seki, Hiroyuki Sugimori.

**Formal analysis:** Takaaki Yoshimura, Shoki Kogame, Kazuya Seki, Hiroyuki Sugimori, Fumi Kato, Kohsuke Kudo.

**Funding acquisition:** Takaaki Yoshimura.

**Investigation:** Takaaki Yoshimura, Hiroyuki Sugimori.

**Methodology:** Takaaki Yoshimura, Shoki Kogame, Kazuya Seki, Hiroyuki Sugimori.

**Project administration:** Takaaki Yoshimura.

**Resources:** Takaaki Yoshimura.

**Software:** Hiroyuki Sugimori.

**Supervision:** Hiroyuki Sugimori, Hiroko Yamashina, Yusuke Nomura, Fumi Kato, Kohsuke Kudo, Shinichi Shimizu, Hidefumi Aoyama.

**Writing – original draft:** Takaaki Yoshimura, Kentaro Nishioka, Takayuki Hashimoto, Takashi Mori, Shoki Kogame, Kazuya Seki, Hiroyuki Sugimori, Hiroko Yamashina, Yusuke Nomura, Fumi Kato, Kohsuke Kudo, Shinichi Shimizu, Hidefumi Aoyama.

**Writing – review & editing:** Takaaki Yoshimura, Kentaro Nishioka, Takayuki Hashimoto, Takashi Mori, Shoki Kogame, Kazuya Seki, Hiroyuki Sugimori, Hiroko Yamashina, Yusuke Nomura, Fumi Kato, Kohsuke Kudo, Shinichi Shimizu, Hidefumi Aoyama.

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
