## [Decision Letter · Decision Letter 0]

22 Jul 2022

PONE-D-22-10629Prostatic urinary tract visualization with super-resolution deep learning modelsPLOS ONE

Dear Dr. Hashimoto,

Thank you for submitting your manuscript to PLOS ONE. After careful consideration, we feel that it has merit but does not fully meet PLOS ONE’s publication criteria as it currently stands. Therefore, we invite you to submit a revised version of the manuscript that addresses the points raised during the review process.

The manuscript is well written. However, The first reviewer finds the manuscript lacking important details, such as details and figures that show the performance of the method, not just one image. Also, please include previous works.

We look forward to receiving your revised manuscript.

Kind regards,

Haydar Celik, PhD

Academic Editor

PLOS ONE

Journal Requirements:

4. Please ensure that you refer to Figure 1 in your text as, if accepted, production will need this reference to link the reader to the figure.

Reviewers' comments:

Reviewer's Responses to Questions

**Comments to the Author**

1. Is the manuscript technically sound, and do the data support the conclusions?

Reviewer #1: Partly

Reviewer #2: Yes

2. Has the statistical analysis been performed appropriately and rigorously? 

Reviewer #1: Yes

Reviewer #2: Yes

3. Have the authors made all data underlying the findings in their manuscript fully available?

Reviewer #1: No

Reviewer #2: No

4. Is the manuscript presented in an intelligible fashion and written in standard English?

Reviewer #1: Yes

Reviewer #2: Yes

5. Review Comments to the Author

Reviewer #1: The work performs an investigation of super-resolution models for improving prostatic urinary tract visualization in PU-MRI. Four deep learning models were investigated and compared against each other. Networks are pre-trained on natural scene images. Qualitative and quantitative assessment identified the RDN as the best candidate for the application.

The manuscript is well written and depicts the major steps in this work. However, several details are not provided, references to related works are missing and not discussed, and only a single image is shown to infer performance. I have detailed out my comments below:

1. Please cite and discuss related works on super-resolution in MRI:

- Küstner, T, Munoz, C, Psenicny, A, et al. Deep-learning based super-resolution for 3D isotropic coronary MR angiography in less than a minute. Magn Reson Med. 2021; 86: 2837– 2852.

- Shi J, Liu Q, Wang C, Zhang Q, Ying S, Xu H. Super-resolution reconstruction of MR image with a novel residual learning network algorithm. Phys Med Biol. 2018;63:085011.

- Pham C-H, Tor-Díez C, Meunier H, et al. Multiscale brain MRI super-resolution using deep 3D convolutional networks. Comput Med Imaging Graph. 2019;77:101647.

- Chen Y, Christodoulou AG, Zhou Z, Shi F, Xie Y, Li D. MRI super-resolution with GAN and 3D multi-level densenet: smaller, faster, and better. arXiv preprint arXiv:200301217 2020.

- Ishida M, Nakayama R, Uno M, et al. Learning-based super-resolution technique significantly improves detection of coronary artery stenoses on 1.5T whole-heart coronary MRA. J Cardiovasc Magn Reson. 2014;16:P218.

2. Previous works have already investigated the recovery of for example the small coronary arteries. In relation to that the authors should motivate and discuss their architectural choices in this work.

3. It is not clear if the authors acquired the PU-MRI in a lower resolution, or if the acquired resolution (not reported) was upscaled by the usage of super-resolution.

4. Please report at least the acquired MR imaging resolution and imaging orientation.

5. Please clarify the low-resolution input to the network (resolution and matrix size) and the high-resolution target. Was a full image inputted or were localized image patches used? Along which dimensions was the image upscaled – only in-plane or also through-plane?

6. Please clarify if a 4x4 upscaling was performed in-plane.

7. L139: Please clarify the performed registration algorithm (rigid, non-rigid, metric, parametrization, …).

8. Why did the authors decide to pre-train the networks on natural scene images? Was any further self-supervision or transfer learning conducted to adapt the networks to the MRI domain? For reference, please refer to [Knoll, F, Hammernik, K, Kobler, E, Pock, T, Recht, MP, Sodickson, DK. Assessment of the generalization of learned image reconstruction and the potential for transfer learning. Magn. Reson. Med. 2018; 81: 116– 128.] describing the risks of using databases of natural scene images. In this regard, was any inpainting or other artifacts originating from the DIV2K database (especially for the GAN-based methods) observed?

9. Please clarify that the networks only operated on the magnitude images instead of the complex-valued MRI data.

10. Was downsampling of the training images performed in image domain? What is done during inference? Are the MR images already acquired in the “input” resolution or are they downsampled beforehand? If so, in image domain or in k-space domain, with the latter being preferred.

11. Why did the authors train the networks for RGB image input instead of changing the input layer to single-channel and training it more closely to the MR domain?

12. Please report the used loss functions, optimizers, learning rates, training epochs, regularization and other hyperparameters used in training.

13. Please report the amount of trainable parameters for each network.

14. Please report the amount of training, validation and test images.

15. Please clarify that the architectures were not further modified and used as proposed in the respective references.

16. Did the authors perform any further ablation studies, e.g. changing to a PatchGAN discriminator, cropped image input or residual super-resolution (operating on full output matrix size as input)?

17. Fig. 2: Please clarify if the bottom left represents the LR input image to the super-resolution networks. Does this example illustrate, the best, average or worst patient?

18. Please provide further super-resolution images of patients to understand the networks’ performances.

19. L272: Why should the image size between the original and SR image differ? A residual super-resolution operates on the output matrix size and does not require this. Or do the authors mean resolution? Please revise.

20. L283-284: Several previous works performed super-resolution on medical images, even trained the networks completely in this domain. This is not a limitation in general and can be overcome depending on the data acquisition and pre-processing.

Reviewer #2: Essentially this well-written and structured paper appears to be aimed at enabling expert contouring of the urethra from normal MRI. The authors have a previous paper on post urination urethra MR imaging. The problem is well motivated by a real clinical problem. Ethics and written consent are addressed. Validation was performed by expert visual comparison with planning CT scans for the same patient (with and without catheter). Scan details were provided in the older paper. The data consisted of 900 2D images from 30 patients. These were converted from the DICOM to PNGs to apply pre-trained SR models.

I liked this paper, but have a few comments:

- Please define SR first time it's used

- The input MRI 2D slices were 310×295 How did the cropping occur from 330x330?

- Was any MRI normalisation or bias field correction applied?

- In terms of image quality comparison, why did the authors only report CW-SSIM? Why not other methods to enable comparison?

- Although time consuming future work could consider contouring the urethra and reporting inter and intra observer differences between experts for RDN and the original MRI.

6. PLOS authors have the option to publish the peer review history of their article (what does this mean?). If published, this will include your full peer review and any attached files.

Reviewer #1: No

Reviewer #2: No

---

## [Author Response · Author response to Decision Letter 0]

7 Sep 2022

We thank all editors and reviewers for their valuable comments, which have helped us to substantially improve our manuscript. Our point-by-point responses to the comments are presented below.

Editor’s comments

1. Thank you for updating your data availability statement. You note that your data are available within the Supporting Information files, but no such files have been included with your submission. At this time we ask that you please upload your minimal data set as a Supporting Information file, or to a public repository such as Figshare or Dryad.

Please also ensure that when you upload your file you include separate captions for your supplementary files at the end of your manuscript.

As soon as you confirm the location of the data underlying your findings, we will be able to proceed with the review of your submission.

Response: We thank the editors for the comment. We have uploaded the supporting information file with our minimal data set of urethra’s visibility score.

2. Please ensure that you refer to Figure 1 in your text as, if accepted, production will need this reference to link the reader to the figure.

Response: We thank the editors for the comment. We have revised the data analysis section, as follows:

Data analysis (Pages 13, Lines 186–188)

Overall, as shown in Fig 1, the inputted in-plane PU-MRI images of these SR deep learning models had a resolution of 310×295 for the LR image and 1240×1180 for the output SR image.

Reviewer #1 comments

The work performs an investigation of super-resolution models for improving prostatic urinary tract visualization in PU-MRI. Four deep learning models were investigated and

compared against each other. Networks are pre-trained on natural scene images. Qualitative and quantitative assessment identified the RDN as the best candidate for the application.

The manuscript is well written and depicts the major steps in this work. However, several details are not provided, references to related works are missing and not discussed, and only a single image is shown to infer performance. I have detailed out my comments below:

 Please cite and discuss related works on super-resolution in MRI:

 Küstner, T, Munoz, C, Psenicny, A, et al. Deep-learning based super-resolution for 3D isotropic coronary MR angiography in less than a minute. Magn Reson Med. 2021; 86: 2837– 2852.

 Shi J, Liu Q, Wang C, Zhang Q, Ying S, Xu H. Super-resolution reconstruction of MR image with a novel residual learning network algorithm. Phys Med Biol. 2018;63:085011.

 Pham C-H, Tor-Díez C, Meunier H, et al. Multiscale brain MRI super-resolution using deep 3D convolutional networks. Comput Med Imaging Graph. 2019;77:101647.

 Chen Y, Christodoulou AG, Zhou Z, Shi F, Xie Y, Li D. MRI super-resolution with GAN and 3D multi-level densenet: smaller, faster, and better. arXiv preprint arXiv:200301217 2020.

 Ishida M, Nakayama R, Uno M, et al. Learning-based super-resolution technique significantly improves detection of coronary artery stenoses on 1.5T whole-heart coronary MRA. J Cardiovasc Magn Reson. 2014;16:P218.

Previous works have already investigated the recovery of for example the small coronary arteries. In relation to that the authors should motivate and discuss their architectural choices in this work.

Response: We thank the reviewer for the comment. We revised the Introduction section and incorporated new references, as follows:

Introduction (Pages 6–7, Lines 78–95)

High-resolution (HR) MRI provides detailed anatomical information. However, HR-MRI is typically characterized by long scan time, small spatial coverage, and low signal-to-noise ratio. Currently, super-resolution (SR) technique aims to generate a HR image from its degraded low-resolution (LR) image using computer vision tasks. SR with deep learning has demonstrated significant advantages over conventional upscaling methods, such as bicubic interpolation in computer vision [9]. In SR with deep learning, learning methods are classified into two types. Dong et al. first demonstrated the training of an SR convolutional neural network (CNN) using supervised learning technique [10]. There are different types of CNN architectures; for example, Shi et al. demonstrated the SR reconstruction of MRI with a novel residual learning network algorithm [11]. The other type of learning technique is unsupervised learning, such as generative adversarial networks (GAN). These deep learning models provide high-quality, natural-looking images. Sood et al. demonstrated that SR using GAN (SRGAN) could be used for SR imaging in prostate MRI [12]. Also, Chen et al. demonstrated that 3D CNN architecture with GAN-guided training provides sharp SR comparable with the referenced HR images [13]. These SR reconstruction techniques have been applied in brain MRI [14] and MR angiography (MRA) [15][16]. Whole-heart coronary MRA permits non-invasive detection of coronary artery disease.

 It is not clear if the authors acquired the PU-MRI in a lower resolution, or if the acquired resolution (not reported) was upscaled by the usage of super-resolution.　Please report at least the acquired MR imaging resolution and imaging orientation.

Response: The details of the image acquisition have been reported in previous studies. We revised the acquired MR imaging resolution and image orientation in Image acquisition section, as follows:

Image acquisition (Page 11, Lines 146–153)

Briefly, PU-MRI was performed using a non-contrast HR two-dimensional T2-weighted turbo spin-echo imaging sequence within a few minutes after urination. The acquisition parameters for PU-MRI were as follows: resolution = 320×320 matrix, voxel size = 0.5×0.5×2.0 mm3, field of view = 160×160 mm2, slices = 30, effective time echo = 80 ms, repetition time = 5093 ms, TSE factor = 9, SENSE P reduction factor = 1.4, gap = 0 mm, and acquisition direction = axial.

 Please clarify the low-resolution input to the network (resolution and matrix size) and the high-resolution target. Was a full image inputted or were localized image patches used? Along which dimensions was the image upscaled – only in-plane or also through-plane? Please clarify if a 4x4 upscaling was performed in-plane.

Response: We thank the reviewer for the comment. We have revised the data analysis section, as follows:

Data analysis (Page 13, Lines 183–189)

Next, to leverage the existing SR deep learning models, the imputed PU-MRI images of these SR deep learning models were cropped at the edge from the original 320×320 resolution to 310×295 resolution by specifying the width and height parameters without changing the image center. Overall, the inputted in-plane PU-MRI images of these SR deep learning models had a resolution of 310×295 for the LR image and 1240×1180 for the output SR image.

 L139: Please clarify the performed registration algorithm (rigid, non-rigid, metric, parametrization, …).

Response: We thank the reviewer for the comment. We have revised the Image acquisition section, as follows:

Image acquisition (Page 11, Lines 151–153)

All acquired images were co-registered with the CT image without using a urethral catheter on MIM ver.7.0.4 (MIM Software, Inc., Cleveland, OH), based on the inserted fiducial markers with rigid registration algorithm.

 Why did the authors decide to pre-train the networks on natural scene images? Was any further self-supervision or transfer learning conducted to adapt the networks to the MRI domain? For reference, please refer to [Knoll, F, Hammernik, K, Kobler, E, Pock, T, Recht, MP, Sodickson, DK. Assessment of the generalization of learned image reconstruction and the potential for transfer learning. Magn. Reson. Med. 2018; 81: 116– 128.] describing the risks of using databases of natural scene images. In this regard, was any inpainting or other artifacts originating from the DIV2K database (especially for the GAN-based methods) observed?

Response: We are grateful for the comment. Although it is possible to obtain higher resolution MRI images, the image quality is compromised by the effect of intestinal gas or internal motion due to the extension of the imaging time, especially in pelvis imaging. Thus, large amounts of true high-resolution training data are often challenging to obtain in clinical practice. Also, the purpose of this study was to evaluate the effect of transfer learning in medical images using existing in super-resolution models for natural images (RGB images), which previously acquired the state-of-the-art (SOTA) performance. Thus, we used the DIV2K database for the purpose of this study. Certainly, as you suggested, there are risks of using databases of natural scene images for transfer learning in medical images. We have revised the limitations paragraph of the Discussion, as follows:

Discussion (Page 22, Lines 313–323)

Knoll et al. decried the risk of using a database of natural scene images for transfer learning in medical images [34]. However, collecting large amounts of data for training is usually expensive in medical imaging, and it is impossible to acquire HR ground truth images in some cases. Especially, it is difficult to prepare a data set specialized for abdominal MRI. Generally, it takes a long time to acquire HR images using MRI. Indeed, the effect of motion artifacts is increased because of internal organ motion during the long acquisition time. In several previous studies, medical image SR was used to train networks by data acquisition and pre-processing [11-13]. Thus, in future studies, we need to increase the urethra visibility with SR deep learning models in the training dataset through data acquisition and pre-processing.

 Please clarify that the networks only operated on the magnitude images instead of the complex-valued MRI data.

Response: We thank the reviewer for the comment. We have added this information to the Image acquisition section, as follows:

Image acquisition (Page 13, Line 188-189)

In this study, all networks operated on only the magnitude images instead of the complex-valued MRI data.

 Was downsampling of the training images performed in image domain? What is done during inference? Are the MR images already acquired in the “input” resolution or are they downsampled beforehand? If so, in image domain or in k-space domain, with the latter being preferred.

Response: We thank the reviewer for the comment. In the training of SR deep learning models, downsampling was performed in image domain. The inputted PU-MRI image was obtained from the device; the inputted image was in magnitude image domain, not k-space domain, and downsampling was performed before input.

 Why did the authors train the networks for RGB image input instead of changing the input layer to single-channel and training it more closely to the MR domain?

Response: We thank the reviewer for the comment. The purpose of this study was to evaluate the effect of transfer learning in medical images using super-resolution models for natural images (RGB images), which previously acquired the state-of-the-art (SOTA) performance. Therefore, the model architecture in this study has three channels corresponding to RGB, and the model architecture was not changed to one channel. We have revised the Methods section, as follows:

Super-resolution deep learning model (Page 13, Line 175–176)

In this study, these architectures were not modified further and were used as proposed in the respective references [21, 23, 25, 26].

 Please report the used loss functions, optimizers, learning rates, training epochs, regularization and other hyperparameters used in training. Please report the amount of trainable parameters for each network. Please report the amount of training, validation and test images.

Response: We thank the reviewer for this comment. We have add a paragraph in the Methods section, as follows:

Super-resolution deep learning model (Pages 12–13, Lines 170–175)

For training, the DIV2K dataset comprises 800 training images, 100 validation images, and bicubic downsampled images. In all SR deep learning models, we train with Adam optimizer [27] by setting β_1=0.9, β_2=0.999, and ϵ=10^(-8). We set the number of epochs as 50, batch size as 16, and learning rate as 10^(-4). The loss functions for the SR deep learning models were L1 loss for EDSR, WDSR, and RDN and VGG54 content loss for SRGAN. 

 Please clarify that the architectures were not further modified and used as proposed in the respective references.

Response: We are grateful for the comment. We have added the following to the Methods section:

Super-resolution deep learning model (Page 13, Lines 175–176)

In this study, these architectures were not modified further and were used as proposed in the respective references [21, 23, 25, 26].

 Did the authors perform any further ablation studies, e.g. changing to a PatchGAN discriminator, cropped image input or residual super-resolution (operating on full output matrix size as input)?

Response: We are grateful for the comment. We did not perform any further ablation studies.

 Fig. 2: Please clarify if the bottom left represents the LR input image to the super-resolution networks. Does this example illustrate, the best, average or worst patient? Please provide further super-resolution images of patients to understand the networks’ performances.

Response: According to your comment, we have selected the best and worst cases in the subjective evaluation. We have modified Fig. 2 as follows:

Fig 2. (Pages 18–19, Lines 257–262)

Fig 2. Representative images identifying the prostatic urinary tract in the axial direction for each SR image.

The above representative images show the best case (No. 15), while the bottom images show the worst case (No. 16) in the subjective evaluation. On the left side of the PU-MRI image is the input image for each SR deep learning model, while on the right side of the PU-MRI image is the enlarged image of the prostate gland and urethra area. The output SR images (B~E) have the same matrix as the magnified PU-MRI image (bottom left). Yellow arrow shows the prostatic urinary tract. 

PU-MRI, post urination magnetic resonance imaging; EDSR, enhanced deep super resolution network; WDSR, wide activation for efficient and accurate image super resolution network; SRGAN, photo-realistic single image super resolution using a generative adversarial network; RDN, residual dense network.

 L272: Why should the image size between the original and SR image differ? A residual super-resolution operates on the output matrix size and does not require this. Or do the authors mean resolution? Please revise.

Response: We thank the reviewer for the comment. We have revised the Discussion as follows:

Discussion (Page 21, Lines 296–303)

The peak signal-to-noise ratio (PSNR) or SSIM has been used for the evaluation of various SR models [17, 19, 25, 26], and these indicators can be used to compare images of the same matrix size. To improve the visibility of the prostatic urinary tract for USRT using the SR deep learning model, we performed four times upsampling of the PU-MRI image by SR deep learning models. As a result, the image matrix size between the original PU-MRI and SR images differed, and PSNR and SSIM were not used as indicators in this study. Therefore, we used CW-SSIM for the objective evaluation of the original images of PU-MRI and SR images.

 L283-284: Several previous works performed super-resolution on medical images, even trained the networks completely in this domain. This is not a limitation in general and can be overcome depending on the data acquisition and pre-processing.

Response: We are grateful for the comment. We have revised the Discussion, as follows:

Discussion (Page 22, Lines 313–323)

Knoll et al. decried the risk of using a database of natural scene images for transfer learning in medical images [34]. However, collecting large amounts of data for training is usually expensive in medical imaging, and it is impossible to acquire HR ground truth images in some cases. Especially, it is difficult to prepare a data set specialized for abdominal MRI. Generally, it takes a long time to acquire HR images using MRI. Indeed, the effect of motion artifacts is increased because of internal organ motion during the long acquisition time. In several previous studies, medical image SR was used to train networks by data acquisition and pre-processing [11-13]. Thus, in future studies, we need to increase the urethra visibility with SR deep learning models in the training dataset through data acquisition and pre-processing.

Reviewer #2 comments

Essentially this well-written and structured paper appears to be aimed at enabling expert contouring of the urethra from normal MRI. The authors have a previous paper on post urination urethra MR imaging. The problem is well motivated by a real clinical problem. Ethics and written consent are addressed. Validation was performed by expert visual comparison with planning CT scans for the same patient (with and without catheter). Scan details were provided in the older paper. The data consisted of 900 2D images from 30 patients. These were converted from the DICOM to PNGs to apply pre-trained SR models. I liked this paper, but have a few comments:

 Please define SR first time it's used

Response: We are grateful for the comment. We have defined SR at first use:

Abstract (Page 4, Lines 45–47)

This study investigated whether the combination of PU-MRI and super-resolution (SR) deep learning models improves the visibility of the prostatic urinary tract.

 The input MRI 2D slices were 310×295 How did the cropping occur from 330x330?

Response: We thank the reviewer for the comment. We cropped the edge of the original PU-MRI image of 320×320 resolution to the input MRI 2D image of 310×295 resolution by specifying the width and height parameters without changing the image center. We have revised the Data analysis, as follows:

Data analysis (Page 13, Lines 183–189)

Next, to leverage the existing SR deep learning models, the imputed PU-MRI images of these SR deep learning models were cropped at the edge from the original 320×320 resolution to 310×295 resolution by specifying the width and height parameters without changing the image center. Overall, the inputted in-plane PU-MRI images of these SR deep learning models had a resolution of 310×295 for the LR image and 1240×1180 for the output SR image.

 Was any MRI normalisation or bias field correction applied?

Response: We thank the reviewer for the comment. In this study, MRI normalization, such as B0 or B1 corrections, which was clinically used, was applied in all images.

 In terms of image quality comparison, why did the authors only report CW-SSIM? Why not other methods to enable comparison? Although time consuming future work could consider contouring the urethra and reporting inter and intra observer differences between experts for RDN and the original MRI. 

Response: We thank the reviewer for the comment. The purpose of this study was to evaluate the effect of transfer learning in medical images using existing super-resolution models for natural images (RGB images), which previously acquired the state-of-the-art (SOTA) performance. We evaluated the use of SR deep learning models for increasing the visibility of urethra for USRT. Our results suggest that SR deep learning may improve the urethra visibility. In a further study, as you suggested, we plan to evaluate the inter- and intra-observer differences of urethra contouring in USRT with the application of more optimized SR deep learning model for urethra visualization after overcoming the limitations of this study.

---

## [Decision Letter · Decision Letter 1]

9 Nov 2022

PONE-D-22-10629R1Prostatic urinary tract visualization with super-resolution deep learning modelsPLOS ONE

Dear Dr. Hashimoto,

Thank you for submitting your manuscript to PLOS ONE. After careful consideration, we feel that it has merit but does not fully meet PLOS ONE’s publication criteria as it currently stands. Therefore, we invite you to submit a revised version of the manuscript that addresses the points raised during the review process.

Thanks to authors for their effort. One of the reviewers asks for minor revision. Please respond throughly. 

We look forward to receiving your revised manuscript.

Kind regards,

Haydar Celik, PhD

Academic Editor

PLOS ONE

Journal Requirements:

Reviewers' comments:

Reviewer's Responses to Questions

**Comments to the Author**

1. If the authors have adequately addressed your comments raised in a previous round of review and you feel that this manuscript is now acceptable for publication, you may indicate that here to bypass the “Comments to the Author” section, enter your conflict of interest statement in the “Confidential to Editor” section, and submit your "Accept" recommendation.

Reviewer #1: (No Response)

Reviewer #2: All comments have been addressed

2. Is the manuscript technically sound, and do the data support the conclusions?

Reviewer #1: Yes

Reviewer #2: Yes

3. Has the statistical analysis been performed appropriately and rigorously? 

Reviewer #1: Yes

Reviewer #2: Yes

4. Have the authors made all data underlying the findings in their manuscript fully available?

Reviewer #1: Yes

Reviewer #2: Yes

5. Is the manuscript presented in an intelligible fashion and written in standard English?

Reviewer #1: Yes

Reviewer #2: Yes

6. Review Comments to the Author

Reviewer #1: I would like to thank the authors for thoroughly addressing my comments and providing further insights. I have a few minor comments remaining:

1. Please clarify in the manuscript that this is a pure retrospective study and that therefore for inference HR images were downsampled in the image domain.

2. Please report the amount of trainable parameters for each network.

3. Please report the amount of test images used for further analysis.

4. L314: Please correct typo “described”

Reviewer #2: Thank you for addressing my comments.

7. PLOS authors have the option to publish the peer review history of their article (what does this mean?). If published, this will include your full peer review and any attached files.

Reviewer #1: No

Reviewer #2: No

---

## [Author Response · Author response to Decision Letter 1]

23 Nov 2022

We would like to thank all of the editors and reviewers for their valuable comments, which have helped us to substantially improve our manuscript. Our point-by-point responses to the comments are presented below.

Reviewer #1 comments

I would like to thank the authors for thoroughly addressing my comments and providing further insights. I have a few minor comments remaining:

 Please clarify in the manuscript that this is a pure retrospective study and that therefore for inference HR images were downsampled in the image domain.

Response: We thank the reviewer for the comment. We have revised the text as follows.

Patients (Pages 8, Lines 115–118)

This retrospective study included a total of 900 images of 30 patients with prostate cancer who had previously undergone real-time-image gated spot-scanning proton therapy (RGPT) using three inserted fiducial markers at our institution from October 2019 to October 2020.

Super-resolution deep learning model (Pages 12, Lines 163–164)

Each LR image was obtained from the HR DIV2K image by bicubic downscaling in the image domain.

 Please report the amount of trainable parameters for each network.

Response: We thank the reviewer for the comment. We have revised the text as follows.

Image acquisition (Page 12–13, Lines 170–178)

The DIV2K dataset comprises 800 training images, 100 validation images, 100 test images, and bicubic downsampled images. In all SR deep learning models, we train with Adam optimizer [27] by setting β_1=0.9, β_2=0.999, and ϵ=10^(-8). The number of epochs was 50; batch size, 16; and learning rate,10^(-4). The loss functions for the SR deep learning models were L1 loss for EDSR, WDSR, and RDN; and VGG54 content loss for SRGAN. The trainable parameters for each network were 43 M, 0.62 M, 1.55 M, and 22 M for EDSR, WDSR, SRGAN, and RDN, respectively. In this study, these architectures were not modified further and were used as proposed in the respective references [21, 23, 25, 26]. For testing, we use 900 PU-MRI images of 30 patients.

 Please report the amount of test images used for further analysis.

Response: We thank the reviewer for the comment. We have included the information per your comment as follows.

Image acquisition (Page 12–13, Lines 170–178)

The DIV2K dataset comprises 800 training images, 100 validation images, 100 test images, and bicubic downsampled images. In all SR deep learning models, we train with Adam optimizer [27] by setting β_1=0.9, β_2=0.999, and ϵ=10^(-8). The number of epochs was 50; batch size, 16; and learning rate,10^(-4). The loss functions for the SR deep learning models were L1 loss for EDSR, WDSR, and RDN; and VGG54 content loss for SRGAN. The trainable parameters for each network were 43 M, 0.62 M, 1.55 M, and 22 M for EDSR, WDSR, SRGAN, and RDN, respectively. In this study, these architectures were not modified further and were used as proposed in the respective references [21, 23, 25, 26]. For testing, we use 900 PU-MRI images of 30 patients.

.

 L314: Please correct typo “described”

Response: We thank the reviewer for the comment. We have revised the sentence as follows.

Image acquisition (Page 22, Line 317)

Knoll et al. described the risk of using a database of natural scene images for transfer learning in medical images [34].

---

## [Decision Letter · Decision Letter 2]

20 Dec 2022

Prostatic urinary tract visualization with super-resolution deep learning models

PONE-D-22-10629R2

Dear Dr. Hashimoto,

We’re pleased to inform you that your manuscript has been judged scientifically suitable for publication and will be formally accepted for publication once it meets all outstanding technical requirements.

Kind regards,

Haydar Celik, PhD

Academic Editor

PLOS ONE

Additional Editor Comments (optional):

Reviewers' comments:

Reviewer's Responses to Questions

**Comments to the Author**

1. If the authors have adequately addressed your comments raised in a previous round of review and you feel that this manuscript is now acceptable for publication, you may indicate that here to bypass the “Comments to the Author” section, enter your conflict of interest statement in the “Confidential to Editor” section, and submit your "Accept" recommendation.

Reviewer #1: All comments have been addressed

Reviewer #2: (No Response)

2. Is the manuscript technically sound, and do the data support the conclusions?

Reviewer #1: Yes

Reviewer #2: Yes

3. Has the statistical analysis been performed appropriately and rigorously? 

Reviewer #1: Yes

Reviewer #2: Yes

4. Have the authors made all data underlying the findings in their manuscript fully available?

Reviewer #1: Yes

Reviewer #2: (No Response)

5. Is the manuscript presented in an intelligible fashion and written in standard English?

Reviewer #1: Yes

Reviewer #2: Yes

6. Review Comments to the Author

Reviewer #1: Thank you for addressing my remaining comments and adding these information into the manuscript. I have no further remarks.

Reviewer #2: (No Response)

7. PLOS authors have the option to publish the peer review history of their article (what does this mean?). If published, this will include your full peer review and any attached files.

Reviewer #1: No

Reviewer #2: No

---

## [Editor Report · Acceptance letter]

27 Dec 2022

PONE-D-22-10629R2 

Prostatic urinary tract visualization with super-resolution deep learning models 

Dear Dr. Hashimoto:

I'm pleased to inform you that your manuscript has been deemed suitable for publication in PLOS ONE. Congratulations! Your manuscript is now with our production department. 

Kind regards, 

on behalf of

Dr. Haydar Celik 

Academic Editor

PLOS ONE